# Genetic Diversity for Agronomic Traits and Phytochemical Compounds in Coloured Naked Barley Lines

**DOI:** 10.3390/plants10081575

**Published:** 2021-07-30

**Authors:** Anna Iannucci, Serafino Suriano, Pasquale Codianni

**Affiliations:** Research Centre for Cereal and Industrial Crops (CREA-CI), Council for Agricultural Research and Economics, 71121 Foggia, Italy; serafino.suriano@crea.gov.it (S.S.); codianni.pasquale@gmail.com (P.C.)

**Keywords:** agronomic traits, breeding, coloured seed, genetic diversity, phytochemicals

## Abstract

Interest of breeders is increasing toward the development of new barley cultivars with functional properties and adapted for food uses. A barley breeding program was initiated to develop germplasm with naked and coloured grains rich in bioactive compounds In the present study, a set of 16 F8 recombinant inbred lines (RILs) derived from the following four parental lines: 2005FG, K4-31, L94 and Priora, were evaluated in the experimental trials in Foggia (Italy) during the 2017–2019 growing seasons with the aims to assess the genetic variability for desired traits and identify superior genotypes. Lines were characterised for agronomic traits (earliness, plant height, seed yield, specific weight, 1000-seed weight) and biochemical compounds accumulation (phenolics, anthocyanins, flavonoids, carotenoids, β-glucans, proteins, antioxidant activity). A high heritability and selection response were observed for most of the biochemical compounds. The grain yield showed high significant positive genetic and phenotypic correlations (*p* < 0.05) with phenols and antioxidant activity. Cluster analysis grouped the genotypes into three groups. The barley RIL lines L1997, L3005, L3007 and L3009 were superior for more than four traits including seed yield and antioxidant compounds. These genotypes may serve as potential sources of nutraceuticals for healthy food and in breeding programs. In the present study, the new barley genotypes with naked and coloured grains have been selected without compromising their productivity.

## 1. Introduction

Barley (*Hordeum vulgare* L.) is a diploid (2n = 14) species belonging to the *Poaceae* family. Due to its adaptability to different types of climates, soil and ease of growing, barley is cultivated both in highly productive agricultural systems and in marginal and subsistence environments [1]. It is one of the most widespread crops in the world, ranked fourth after wheat, rice and maize [2]. Approximately 65% of cultivated barley is used for animal feed, 33% for malting, whereas only 2% is used directly for human consumption as a component of various food products [3]. Recently, barley has gained renewed attention by the increasing demand for traditional and healthy products; therefore, breeding programs aimed at increasing the content of nutritional substances of barley kernels are ongoing.

Cereals and their derivatives are the most important foods in the human diet as a source of macronutrients providing significant quantities of carbohydrates (thus, energy), proteins and micronutrients. Today, whole barley grains have been recognised as a functional food, because their high dietary fibre levels (mainly β-glucans), mineral and phytochemicals can provide a beneficial effect on the health of the consumer and decrease the risk of various diseases [4]. Epidemiological studies have associated the regular consumption of whole barley flour with its potential to reduce the risk of hyperlipidemia, diabetes and other pathological conditions [5]. The enrichment of foods by the addition of a high-fibre cereal grain containing β-glucans is one way to increase fibre intake.

β-glucans are a family of non-starch polysaccharides that have the property to form high viscosity solutions, lowering both cholesterol and serum triglycerides [6]. Whole grain barley also contains phytochemicals (non-nutrient components) such as phenolics, a heterogeneous group of compounds, which include flavonoids, phenolic acids, tannins, stilbenes, anthocyanins, xanthenes and lignans [7]. Phytochemicals in barley may exist in free, conjugated or bound forms and exhibit strong antioxidant activities and ability to scavenge free radicals and metals, and to inhibit lipid peroxidation [8].

The seeds of some cereals, including barley, may have a yellow, blue or purple colour caused by an accumulation of compounds, such as polyphenols, tannins, anthocyanins and carotenoids [9], which are primarily concentrated in the outer layers of the kernel. As the genes regulating the blue, black and purple colour traits are distinct, it is possible to increase specific biochemical compounds [9]. Therefore, grains of coloured barley seem to be a promising ingredient for the development of functional food [4]. However, to date, few coloured cultivars adapted to the Mediterranean environments have been developed.

Traditionally, barley cultivars have a caryopsis firmly adhered to the hull and this is a disadvantage from an industrial point of view due to the need for a dehulling process and grain storage. These factors could be overcome by the commercial exploitation of hull-less or ‘naked’ cultivars [10]. Naked barley is a form of domesticated barley with an easier-to-remove hull. The appearance of the naked characteristic of the caryopsis is due to a single recessive gene (*nud* for *nudum*) that regulates the lack of production of one cementing substance by the caryopsis after pollination, responsible for the adhesion of the glumellae to the seminal teguments [11]. Naked barley is preferred for human consumption and has been investigated for multiple food applications as it can be used as it is. Indeed, pearling reduces the food value because it also removes, in addition to integuments, part of the aleurone and embryo. A whole grain contains generally more protein and starch and total and soluble β-glucans [5]. However, hull-less barley is not grown widely due to its low productivity and weak straw [12]. Nowadays, very few naked barley cultivars have been developed for modern European and North American agricultural systems [13]. The introgression of the naked phenotype is currently an interesting goal of breeding to develop naked multi-use barley lines for conventional and organic systems [14].

Barley’s genetic improvement is aimed almost exclusively at developing cultivars with high and stable grain yield. Therefore, it is very important for breeders to develop efficient selection programs, through targeted genetic crosses, for naked and coloured barley lines rich in antioxidant properties and bioactive compounds. The success of such breeding programs depends primarily on the assessment of the important characters and of the genetic diversity of the existing germplasm [15]. Although considerable genetic variability has been reported in the agronomic traits and nutritional composition of this species [1], heritability estimates and the expected gain from selection are still little investigated. Recently, Suriano et al. [16] characterised some Italian coloured naked barley lines with a high total polyphenols content together with a high antioxidant capacity. However, the main purpose in any breeding program is to produce a variety that will be adopted by growers. Therefore, it is necessary to assess the agronomic performance of promising breeding materials.

In the present work, agronomic evaluation was carried out on advanced breeding lines of the coloured naked barley selected for qualitative characteristics and on their parents. In particular, the aims of this study were (i) to assess the agronomic performance and the genetic parameters of the lines; (ii) to know the relationship between seed yield and seed qualitative traits and (iii) to identify the superior genotypes that can be used in the production/transformation chains of functional foods.

## 2. Results

Based on 15-year average meteorological data (2002–2016), the climate through the plant growth season (November–June) was characterised by mean precipitation of 428 ± 34 mm yr^−1^ and mean minimum and maximum temperatures of 7.1 ± 2.5 °C and 19.1 ± 2.3 °C, respectively (Table 1). The maximum and minimum temperatures were similar and lower, respectively than the 15-year average for the 3 years of the study. During the 2016–17 and 2017–18 growing seasons, we recorded lower rainfall than the long-term means, whereas the last year was characterised by very intense rainfall. However, all the years showed very heavy rainfall (11.5 mm, on average) at the end of the growing season (May–June) compared to the 15-year average.

Significant differences (*p* < 0.0001) were observed among the Genotype (G), Year (Y) and G × Y interaction effects for all of the agronomic traits examined (Appendix A). Significant differences (*p* < 0.001) were also recorded among the genotypes for all the chemical compounds measured (Appendix A).

The extent of the variation was assessed by an estimation of the genetic parameters (Table 2). The data indicated wide ranges of variation with CV (%) values higher than 10% for most traits, with the exception of plant height (PH), specific weight (SW), 1000-seed weight (TSW) and total proteins (TPro). For most traits, a large portion of the phenotypic variance (σ^2^_p_), i.e., more than 67%, was accounted for by the genetic component (σ^2^_g_). For seed yield (SY), σ^2^_g_ accounted for only 18% of the σ^2^_p_. The phenotypic coefficient of variation (PCV) was slightly higher than the genotypic coefficient of variation (GCV) for almost all the traits. In general, the estimates of both PCV and GCV showed a wide range of variations (3.7 and 3.6% in SW to 42.8 and 42.6% in total anthocyanins (TAnt), respectively). The PCV and GCV were high (>20%) for total carotenoids (TCarot) and Tant; intermediate (10–20%) for days to heading (HT), β-glucans (β-glu), total polyphenols (TPh), total proanthocyanidins (TPAnt), total flavonoids (TFlav) and antioxidant capacity (DPPH and TEAC) and low (<10%) for the other traits. Broad-sense heritability for the whole collection was high and ranged from 67.6% (TFlav) to 98.9% (TAnt), only SY showed a lower heritability value (15.6%). More than half of the traits (79%) exhibited heritability in a broad sense greater than 80%. Selection gains higher than 10% would be expected in 11 traits if 5% of the lines were selected. ΔG was particularly high for TAnt (87.3%) and TCarot (54.5%), while it was minimal for SW (7.1%), SY (7.3%) and PH (9.6%). With the exception of the agronomic traits, TPro and TFlav, all the other traits showed both a high heritability and genetic gain.

An analysis of the correlation coefficients was performed to find out the inter-relationship between the pairs of traits studied. Both the genotypic (below diagonal) and phenotypic (above diagonal) pairwise correlation coefficients were presented in Table 3. Seed yield had a negative phenotypic correlation with PH (r_p_ = − 0.59, *p* ≤ 0.01) and a significant positive association with TPh (r_p_ = 0.47, *p* ≤ 0.05), DPPH (r_p_ = 0.52, *p* ≤ 0.05) and TEAC (r_p_ = 0.46, *p* ≤ 0.05) (Table 3). HT was positively correlated with TPant (r_p_ = 0.63, *p* ≤0.01) and negatively correlated with SW (r_p_ = − 0.44, *p* ≤ 0.01) and TAnt (r_p_ = − 0.63, *p* ≤ 0.01). TPh was positively correlated with the antioxidant activity (r_p_ = 0.81 and r_p_ = 0.91, *p* ≤ 0.01, for DPPH and TEAC, respectively). Finally, TCarot showed a negative association with TAnt (r_p_ = − 0.54, *p* ≤ 0.05) and SW (r_p_ = − 0.84, *p* ≤ 0.01). The two methods used to measure the antioxidant activities were highly positively correlated (r_p_ = 0.84, *p* ≤ 0.01). The genetic correlation coefficients were higher than the corresponding phenotypic ones.

The PCA was performed, and the first five axes showed eigenvalues >1 and accounted for more than 80% of the observed phenotypic diversity for the barley genotypes examined (Table 4). The first component (PC1) explained 27.2% of the variation and was positively correlated with TPh (0.42), DPPH (0.43) and TEAC (0.42). The second component (PC2), explaining 22.4% variance, was more positively correlated with HT (0.40) and TPAnt (0.42) traits. 

The third component (PC3) explained 12.3% of the variance and TPro (0.52) was the trait with the largest contribution to the genetic divergence in the genotype set evaluated. TSW (0.67) showed the highest value for the fourth component (PC4) and accounted for 11.5% of the variation in the dataset. Finally, PC5 explained 7.4% of the total variability, and was positively correlated to β-glu (0.60) and TFlav (0.55).

Hierarchical clustering based on agronomic and biochemical traits identified three clusters, i.e., I to III, that included 11, 7 and 2 genotypes, respectively (Figure 1). Based on the means calculated over the genotypes included in each cluster, cluster I showed short and early heading plants with a high seed-specific weight, TAnt and TFlav contents (Appendix A). Cluster II was characterised by taller plants with a high SW, protein and β-glucan contents and high antioxidant activity. Cluster III was composed by two parental genotypes (RIL ‘K4–31’ and cv. ‘Priora’) with later heading plants characterised by a high TSW, TCarot, β-glucans and TPAnt. A heat map was constructed to visualise the genotypes that were superior for one or more traits with respect to the other ones (Figure 1).

Some genotypes with interesting characteristics were found within clusters I and II. Table 5 shows the number of superior genotypes for each trait within each group based on the genotypes with trait values that exceeded the mean plus standard deviation (SD). For HT and PH, the mean minus the SD was used, as these traits are useful for better crop performance under Mediterranean conditions. According to these criteria, the highest number of superior genotypes was 11 for HT and 6 for DPPH, and the lowest number was 1 for both SW and TSW. Seven of the lines showed better performance for at least three different traits. These superior lines showed superior values for SY (+20.05%) and for all the biochemical compounds (from +2.2 to 302.8% for β-glucans and total anthocyanins, respectively). Furthermore, the lines exhibited lower values for HT (–16.9%) and PH (–12.3%). Compared to the parents, line L3005 was 3 days earlier for HT, 13 cm shorter for PH and 20% more productive than SY. The lines L1997, L3005, L3007 and L3009 were superior according to more than four of the traits.

## 3. Discussion

In the present study, the analysis of variance showed that there is considerable variation among barley genotypes for all of the characters under investigation. The genotype × environment interaction was significant for all the agronomic traits indicating the importance of carrying out tests over several years to evaluate the crop performance. We found high mean values for seed yield and specific weight, remarkably lower values for plant height and earliness for some genotypes. High concentrations were found for all the biochemical compounds examined and the wide range recorded is adequate to distinguish the inbred lines using the parameters studied; this facilitates the selection of suitable genotypes for pre-breeding and further evaluation. 

Information on the variation present in a population due to genetic and non-genetic factors provides a basis for the development of efficient breeding methods [17]. We found the genetic variance was the main component of phenotypic variation among genotypes for almost all traits, and very low differences were observed in the estimates of PCV and GCV indicating the existence of enormous intrinsic variability that can remain unaffected by environmental conditions. Therefore, the ranking of genotypes across environments should be relatively constant for these traits. Based on the categorisation used here, a high PCV and GCV were recorded for anthocyanins and carotenoids. Similar results were reported in pigmented rice and durum wheat [18,19]. Low GCV and PCV estimates for seed weight, thousand seed weight and plant height were also reported by Yadav et al. [20].

Although the genotypic coefficient of variation reveals the extent of genetic variability present in the genotypes, it does not allow heritable variation to be assessed. Estimates of heritability and genetic advances are helpful in predicting the response to selection and the expected progress to be achieved through the selection process [21]. Both heritability and genetic advance for carotenoids, phenols, anthocyanins, proanthocyanidins, flavonoids and β-glucans were high to very high. The high genetic advance and heritability suggest the presence of additive gene action for their expression, which is fixable for the next generation [17]. Each of these compounds could, therefore, be used for selection and screening, and few environments may be enough to evaluate these traits for breeding. Indeed, recent studies indicated that many biochemical compounds present in barley grain and other cereals, such as yellow pigments, phenols [22] and carotenoids [23], may be influenced more by the genetic component than by the environmental one.

Our study showed that barley genotypes exhibited significant free radical scavenging activities and the antioxidant compound contents varied among them. According to Chen et al. [24], at the present day, this characteristic is required to be increased in barley grains. Phenolic compounds, in particular phenolic acids, are considered as a major group of compounds that contribute to the antioxidant activity of cereals [25]. Previously, Suriano et al. [16] found a large variability among barley genotypes in total free and total bound phenols and the insoluble bound phenolic acids represented 88.3% of the total phenolic acids. Di Silvestro et al. [26] reported that the soluble fraction of phenolics in wheat cultivars was mainly determined by the environment, while the major genotypic effect was observed for the bound forms present in great amounts in red-grain cultivars. In our set of coloured barley lines, good levels of the other antioxidant compounds such as anthocyanins, flavonoids and proanthocyanidins were detected. As reported by Loskutov and Khlestkina [27], the pigmentation of the grain’s outer coating, attributable to anthocyanins, can be analysed as an important indicator of antioxidant activity. Moreover, also the content of flavonoids and proanthocyanidins in barley grains is proportional to the degree of colour depth shown by the varieties with blue and purple seeds [23,28]. The high genotypic coefficients of variation, together with a high estimate of broad sense heritability and high genetic advance, were observed for carotenoids and anthocyanins. This result suggests that these traits can have a great importance in breeding programs aiming at improving barley for coloured seeds. According to Goudar et al. [12], the β-glucan content of our genetic materials varied from 2.5 to 4.6%. The β-glucan content of barley grain is mainly determined by genotype and less by environmental factors [27,29]. We observed high heritability with moderate genetic advance for grain protein content. This indicates the presence of non-additive gene action; thus, a simple selection procedure in the early segregation generation may not be effective for generating lines with desirable traits for future plant breeding [30].

For the agronomic traits, the lowest heritability was recorded for seed yield in our experiment, perhaps because yield is a character controlled by many genes. Moreover, as reported by Hernandez et al. [14], the progress in selection for this trait is, in general, slow due to the masking effect of the environment and to a narrow germplasm base in barley. Furthermore, according to Yadav et al. [20], we found a high heritability and a moderate genetic advance for the days to heading and thousand seed weight. For barley cultivation, earliness is one of the crucial traits much appreciated by farmers, as it is useful for escaping drought and terminal heat stress and for enhancing yield stability in Mediterranean environments [2]. Finally, a high heritability, along with low genetic advance, was recorded for plant height and specific weight. Similar results were also reported by Mishra et al. [31].

Genotypic correlation coefficients were greater for most of the characters than their corresponding phenotypic correlation coefficient values, indicating the close association of the characters. By utilising the genetic correlations between traits, secondary traits can be used to improve primary ones that have a low heritability or are difficult to measure [32]. A negative association between seed yield and plant height was observed in the present study; therefore, the indirect selection for short plants could increase the grain yield in barley. A decreased plant height was used to reduce the yield loss due to lodging and to increase the harvest index [33]. Positive significant associations were observed between the seed yield and the total phenolic compounds and antioxidant activity. As suggested by Lin et al. [34], phenolic compound content could be used as a biochemical marker to characterise barley genotypes. The concentrations of the total phenols and antioxidant activity were positively correlated; this implies that phenols were the major contributors to the antioxidant capacity. The lack of significant genotypic and phenotypic correlation between the seed yield and the other phytochemical compounds indicates the possibility for the improvement of these nutrients without a significant reduction in seed yield. According to Fernandez-Orozco et al. [35], no significant correlations were found between the phenolics content and the grain properties, such as the thousand kernel weight or the protein content. A negative correlation between total carotenoids and specific weight was also observed in durum wheat [36] and, this suggests trade-offs in breeding for these antioxidant pigments and heavier seeds. Finally, high correlation coefficients were obtained between the DPPH and ABTS antioxidant methods, suggesting that the compounds that can scavenge the DPPH radical in these barley genotypes also scavenged the ABTS radical cation [16].

The measurement and classification of genetic variability between genetic materials are both an important assist in parent selection. For this purpose, PCA and cluster analysis are complementary and valuable statistical tools. While the extent of variability is measured via PCA, the classification of the variability is accomplished via cluster analysis. In this study, the first five PC axes (eigenvalues ≥1.0) represented a cumulative variance of 81% and the major characters responsible for divergence among barley lines were identified. Clustering of genotypes on the basis of their genetic similarity aids in the identification and selection of the best parents for specific breeding programs [2]. The RILs included in this research were grouped into two clusters. Members of group I can be selected as suitable parental materials for hybridisation whenever low size, earliness, seed weight and antioxidant compound content are the breeding objectives. On the contrary, members of group II showed a high content of protein and β-glucan and a high antioxidant activity. Therefore, the members of groups I and II could be used as complementary parents for crossing to obtain a heterotic response. The results indicate that, based on each trait, various RILs exceeded the parents (from 1 for SW and TSW to 11 genotypes for HT). Seven superior lines resulted particularly rich in phytochemical compounds and/or for antioxidant activity, suggesting that they could be chosen to improve the health value of barley end-products. The total phenolics concentrations that ranged from 2691 to 2917 μg (GAE) g^–1^ DM for these barley lines were higher compared to the values reported in previous studies (less than 2210 μg (GAE) g^–1^ DM) [29,37]. The lines L1997, L3005, L3007 and L3009 were superior according to more than four of the traits.

## 4. Materials and Methods

### 4.1. Plant Materials 

A set of 16 F8 recombinant inbred lines (RILs) (namely L1956, L1960, L1966, L1981, L1997, L2000, L2002, L3001, L3002, L3003, L3004, L3005, L3006, L3007, L3008, L3009), as reported by Suriano et al. [16], were obtained using single-seed descent method and developed through three different parental backcross programs between line 2005FG (blue and naked grains), RIL K4-31 (black and hulled grains), Ethiopian landrace line L94 (black and naked grains) and cv. Priora (white and naked grains and rich in β-glucans). The RILs were screened for naked seed, grain colour (according to the descriptors of barley indicated in IPGRI, 1984), seed yield (value > mean + standard deviation) and lodging resistance (value ˂ mean – standard deviation) within a wide segregating population during the 2015–2016 growing season (Appendix A). 

### 4.2. Field Trials

For the evaluation of the agronomic traits, the 16 RILs were grown with their respective parents during three growing seasons (2016–2017, 2017–2018 and 2018–2019) in field experiments at the Research Centre for Cereal and Industrial Crops (CREA-CI) in Foggia (southern Italy) (41°28′N, 15°34′E; 76 m a.s.l.).

The trial was performed on a clay-loam soil (Typic Chromoxerert) with the following characteristics: 36.9% clay, 50.5% silt, 12.5% sand, pH 8 (in H_2_O), 15 mg kg^−1^ available phosphorus (Olsen method), 800 mg kg^−1^ exchangeable potassium (NH_4_Ac), 1.5 g kg^−1^ total nitrogen and 21 g kg^−1^ organic matter (Walkey–Black method). The previous crop was fallow, which was ploughed, hoed and harrowed twice prior to the barley planting. 

The experiments were arranged in a randomised complete block design, with three replications. The plot area was 10 m^2^, with 8 rows that were each 7.5 m long and 0.17 m apart. The experimental blocks were separated by 1.5 m walkways. Sowing was performed with a plot driller at a seeding density of 350 viable seeds m^−2^ on 5 December 2016, 11 December 2017 and 6 December 2018. The fertiliser used at sowing was 18/46 fertiliser (18% elemental nitrogen; 46% P_2_O_5_; by weight) applied at 200 kg ha^−1^, and at plant tillering, NH_4_NO_3_ (26% elemental nitrogen) was applied at 200 kg ha^−1^. Weed control was carried out at the end of tillering, using Manta Gold (fluroxipir, 6.0% (60 g L^−1^) and clopiralid, 2.3% (23.3 g L^−1^); [4-chloro-2-methylphenoxy] acetic acid, 26.7% (266 g L^−1^)) mixed with Axial Pronto (pinoxaden, 6.4% (60 g L^−1^); cloquintocet-mexyl, 1.55% (15 g L^−1^)). Each year, the central six rows of each plot were combine harvested after physiological maturity on 15 June 2017, 25 June 2018 and 13 June 2019. Meteorological data were obtained from an on-site weather station.

The biochemical analyses were performed on samples harvested during the 2015–2016 growing season. Plants were sown on 10 December 2015 and harvested on 14 June 2016. The experiment was conducted under the same conditions previously reported. The barley grains were stored in a cool chamber at 5 °C until analysis. Before analysis, the barley samples were ground using a sample mill (Cyclone; Udy Corp., Fort Collin, CO, USA) equipped with a 0.5-millimetre screen. The chemical compositions were analysed as two subsamples of each grain sample.

### 4.3. Agronomic Traits 

Heading time (HT; days) was recorded as the number of days from April 1 until the ears of 50% of the tillers had emerged from the flag-leaf sheaths by approximately half of their length (i.e., Zadoks scale, growth stage 55; Zadoks et al. [38]). Plant height (PH; cm) was measured from the ground to the tip of the ear (excluding awns), during the early dough development stage (i.e., Zadoks scale, growth stage 83), when the maximum height was achieved, on five main culms per plot. Seed yield (SY; t ha^−1^) was determined and expressed at 13% moisture level. Specific weight (SW; kg hL^−1^), an indication of the density of the grain, was measured on 250-gram samples per plot, using a Schopper Chondrometer equipped with a 1-litre container, without reference to the moisture content. Thousand-grain weight (TSW; g) was calculated as the mean weight of five sets of 200 seeds.

### 4.4. Biochemical Analyses 

The chemical analyses were performed according to the protocols described by Suriano et al. [16] for proteins (TPro), carotenoids (TCarot) and β-glucans (β-glu), and according to Suriano et al. [39] for phenolic compounds and antioxidant activities. The extraction and purification of all the phenolic compounds and for the antioxidant activities from the barley grain were performed according to the method of Ficco et al. [40]. The spectrophotometric determinations were carried out using a double-beam high-performance UV/VIS PC spectrophotometer (Lambda 25; Perkin Elmer SpA). All of the analyses were conducted in triplicate. The methods are summarised here.

Protein content was determined using the Dumas combustion nitrogen method, according to American Association of Cereal Chemists (AACC, 2012) Approved Method 46–30.01 and using FP528 (Leco Corp., Saint Joseph, USA). A factor of 5.7 was used to convert the nitrogen to protein. Data are given as g (100 g^−1^) dry matter (DM).

The reference method for the pigment total carotenoid content determination used was AACC Method 14–50 (AACC, 2013), with minor changes [41]. The total carotenoids content (TCC) of the extracts were calculated directly from the absorbance using the conversion factor of 1.6632 and are expressed as β-carotene (1 mg β-carotene in 100 mL water-saturated 1-butanol has an optical density of 1.6632 in 1-centimetre cuvette at 435-nanometre wavelength).

β-Glucan content was determined following ICC Generic Methods N° 166 (1996) (K-BGLU Megazyme kits). Determination was carried out by spectrophotometric methods using a UV/VIS double-beam high-precision PC spectrophotometer.

The total polyphenols (TPh) for the barley grain were determined using a procedure adapted and modified by Gao et al. [42], using Folin–Ciocalteu reagent. The measurements were quantified according to a standard curve of 0.3–1.0 mmol L^−1^ gallic acid and the total polyphenols are expressed as μg gallic acid equivalents (GAE) g^–1^ DM.

The total proanthocyanidins (TPAnt) were determined according to the modified vanillin assay of Sun et al. [43]. The absorbance was measured at 500 nm after 20 min. The total proanthocyanidins are expressed as μg g^–1^ catechin in comparison with the standard (+)-catechin treated under the same conditions.

The total anthocyanins (TAnt) for the barley grain were evaluated using a colorimetric method [40] with different pH solutions and are expressed as μg Cy3-glc equivalents g^–1^ dry matter (DM).

The total flavonoids (TFlav) were determined according to the method of Zhishen et al. [44]. The absorbance was determined against a blank at 510 nm. (+) Catechin was used for the standard curve (0.05–0.5 mg mL^−1^) and the data are expressed as μg catechin equivalents (CE) g^–1^ DM.

The antioxidant capacity was determined according to the following two different assays: DPPH radical scavenging activity and ABTS radical scavenging activity. The DPPH 2,2-diphenyl-1-picrylhydrazyl radical scavenging capacity of the barley grain extracts was determined according to Brand-Williams et al. [45]. The absorbance was measured at 517 nm. The range of concentrations of Trolox used for the calibration curve was 0–80 lM. The data are expressed as μmol Trolox equivalents (TE) g^–1^ DM. The TEAC Trolox equivalent antioxidant capacity was determined according to the procedure of Fares et al. [46]. The TEAC of the extracts were calculated using a Trolox standard curve on the basis of the percentage inhibition of absorbance at 734 nm and they are expressed as μmol TE g^–1^ DM. The range of concentrations of Trolox used for the calibration curve was 0–60 μM.

### 4.5. Statistical Analysis

Descriptive statistics include mean, standard error (SE), range (min-max) and coefficient of variation (CV, %), which, for the agronomic traits, were calculated over the three growing seasons to describe the variability among the barley lines. 

Analysis of variance (ANOVA) was performed for each trait according to GLM procedure. The homogeneity of the error variance was verified using Bartlett’s tests. The total phenotypic variation of each trait was partitioned into the variance components due to genetic and nongenetic factors, according to Falconer and Mackay [21]. Genotypic variance (σ^2^_g_), phenotypic variance (σ^2^_p_) and coefficients of variation of the genotypes (GCV) and phenotypes (PCV) were estimated. We classified the GCV and PCV values as low (0–10%), moderate (10–20%) and high (>20%) [30].

Broad-sense heritability (h^2^_b_) was estimated as follows:h^2^_b_ = σ^2^_g_/σ^2^_p_ × 100(1)
where σ^2^_g_ is the genotypic variance and σ^2^_p_ is the phenotypic variance. The components of the phenotypic variance were calculated on the basis of the analysis of variance of the expected mean squares. The phenotypic variance was therefore calculated as follows:σ^2^_p_ = σ^2^_g_ + σ^2^_gy_/y + σ^2^_e_/r × y(2)
where σ^2^_gy_ is the genotype × year interaction variance, σ^2^_e_ is the error variance, r is the number of replicates and y is the number of years. According to Crespel et al. [47], years were considered as different environments. For the biochemical compounds the term σ^2^_gy_ was not included in the formula. According to Johnson et al. [30], heritability > 80% is very high, from 60% to 79% is moderately high, from 40% to 59% is medium and <40% is low.

Response to selection (R) was calculated as follows:R = h^2^_b_ × i × √σ^2^_p_(3)
where i is the standardised selection differential at 5% selection intensity (i = 2.063).

Then, the R as a percentage of the mean (ΔG) was calculated to compare the extent of predicted R of the different traits under selection, using the following formula: ΔG (%) = R/μ × 100(4)

According to Johnson et al. [30], the genetic advance as a percent of the mean was categorised into low (<10%), moderate (10–20%) and high (>20%). High and low values are indicative of additive or non-additive gene actions, respectively.

The phenotypic (r_p_) and genotypic (r_g_) correlation coefficients for the different traits for all possible combinations were estimated using variances and covariances, according to the following method described by Falconer and Mackay [21]:r_p_ = cov_p1-2_/√ (σ^2^_p1_ × σ^2^_p2_(5)
r_g_ = cov_g1-2_/√ (σ^2^_g1_ × σ^2^_g2_)(6)
where cov_p1-2_ and cov_g1-2_ are the phenotypic and genotypic covariances between traits 1 and 2, and σ^2^_p1_, σ^2^_p2_ and σ^2^_g1_, σ^2^_g2_ are the phenotypic and genotypic variances of the two traits, respectively.

The relative importance and contribution of each trait to multivariate polymorphism were tested using PCA. PCA was calculated to extract the factorial load of the matrix and also to estimate the number of factors. Principal components with an eigenvalue ≥ 1 were retained for analysis and subjected to hierarchical cluster analysis. Cluster analysis was performed to better understand the degree of divergence and relatedness among the genotypes. A Euclidean distance matrix was established from the PCA values to obtain a relative dendrogram. The genotypes were clustered using Ward’s minimum variance method. To identify the group to which each genotype belonged, the automatic truncation available as part of the software was used.

To determine the significant differences of each trait among the clusters, nested analysis of variance (NANOVA) was performed using the restricted maximum likelihood procedure. Mean comparison was performed by applying the Tukey–Kramer tests, with statistically significant differences determined at the probability level of *p* ≤ 0.05.

The statistical analyses were performed using the JMP software (version 8.0; SAS Institute Inc., Cary, NC, USA).

## 5. Conclusions

Our results indicate that it was possible to obtain, through a breeding program, barley lines with naked and coloured seeds and with good agronomic characteristics and nutritional properties. The barley lines examined are good sources of phenolic compounds, β-glucans and antioxidants. Moreover, the highly positive correlations between the antioxidant capacities and the phenolic compounds indicate that the latter could be the main contributors to antioxidant activity in barley grain. A high genotypic coefficients of variation, coupled with a high estimate of broad sense heritability and a high genetic advance were observed for anthocyanins and carotenoids. This result suggests the importance of these traits in a selection program that targets barley improvement for coloured seed. The lines L1997, L3005, L3007 and L3009 showing a desirable combination of seed yield, phytochemical compounds and for antioxidant activity can be used for improving the existing food barley varieties for these traits. These selections can also represent an important contribution to the development of food products with high nutritional properties.

## Figures and Tables

**Figure 1 plants-10-01575-f001:**
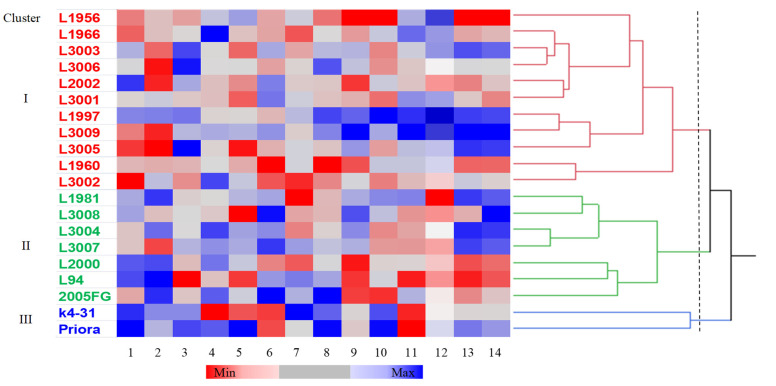
Hierarchical cluster analysis dendrogram (Ward’s method) and pattern map of the values of the agronomic and qualitative traits of 20 barley genotypes. Abbreviations for graph: 1, days to heading from April 1; 2, plant height (cm); 3, seed yield (t ha^−1^); 4, specific weight (kg hL^−1^); 5, 1000−seed weight (g); 6, total proteins (%); 7, total carotenoids (μg β carotene g^−1^ dry matter); 8, β-glucans (%); 9, total polyphenols (μg gallic acid equivalents (GAE) g^−1^ dry matter); 10, total proanthocyanidins (μg catechin equivalents (CE) g^−1^ dry matter); 11, total anthocyanins (μg CE g^−1^ dry matter); 12, total flavonoids (μg gallic acid g^−1^ dry matter); 13, DPPH (μmol of Trolox equivalent g^−1^ dry matter); 14, TEAC (μmol of Trolox equivalent g^−1^ dry matter).

**Table 1 plants-10-01575-t001:** Environmental data at Foggia (Italy) during three growing seasons (2017–2019) as compared to the long-term (15-year) averages.

Month	Maximum Temperature (°C)				Minimum Temperature (°C)				Rainfall (mm)			
	2016–2017	2017–2018	2018–2019	15-Year Average	2016–2017	2017–2018	2018–2019	15-Year Average	2016–2017	2017–2018	2018–2019	15-Year Average
November	20.8	22.8	22.6	18.0	11.8	9.7	13.5	7.6	41.0	15.8	101.2	58.6
December	17.1	16.2	16.8	13.7	7.3	6.1	8.7	4.0	38.4	69.1	78.6	72.2
January	13.1	12.4	13.0	13.1	1.7	3.2	4.4	3.3	3.1	21.5	39.1	61.2
February	8.3	13.9	9.7	13.8	1.1	3.4	1.8	3.2	81.8	31.9	60.2	44.7
March	14.8	10.3	13.6	16.9	5.3	2.3	2.5	4.7	23.2	52.9	29.1	55.2
April	17.6	15.1	17.6	20.8	5.2	5.7	5.6	7.5	8.2	54.3	23.2	52.9
May	18.9	22.6	20.4	25.8	5.5	8.3	8.0	11.4	23.5	6.7	35.4	37.2
June	25.0	25.3	20.2	31.1	10.5	12.9	9.7	15.5	80.0	97.9	90.7	45.7
Mean	17.0	17.3	16.7	19.1	6.1	6.5	6.8	7.1				
Total									299.2	350.1	457.5	427.6

Note: Data derived from on-site weather station at Foggia.

**Table 2 plants-10-01575-t002:** Phenotypic performance and estimates of genetic parameters for the agronomic and qualitative traits of 20 barley genotypes.

Trait	Whole Collection								
	Mean	Range	CV (%)	σ^2^_g_	σ^2^_p_	GCV (%)	PCV (%)	h^2^_b_ (%)	ΔG (%)
HT	12.2	10.0–15.0	11.6	1.6	2.0	10.5	11.4	83.6	19.7
PH	96.8	88.9–107.2	5.8	25.0	30.7	5.2	5.7	81.6	9.6
SY	4.6	3.4–5.4	9.7	0.2	1.1	9.0	22.8	15.6	7.3
SW	79.9	68.0–82.1	3.7	8.0	8.6	3.6	3.7	93.5	7.1
TSW	38.2	34.8–45.5	6.3	4.6	5.6	5.6	6.2	82.1	10.5
TPro	12.8	10.8–14.4	8.0	1.0	1.0	8.0	8.0	99.8	16.4
TCarot	2.3	1.7–4.6	28.2	0.4	0.5	27.9	29.3	90.2	54.5
β-glu	4.1	2.5–4.6	12.3	0.2	0.3	11.9	12.6	90.6	23.4
TPh	2374.8	1929.0–2917.0	11.5	74,143.0	77,141.0	11.5	11.7	96.1	23.2
TPAnt	1336.7	1037.0–1630.0	11.5	22,753.7	25,021.7	11.3	11.8	90.9	22.2
TAnt	81.9	1.0–132.3	44.1	1217.7	1230.8	42.6	42.8	98.9	87.3
TFlav	1041.3	657.0–1421.7	17.5	28,736.6	42,500.8	16.3	19.8	67.6	27.6
DPPH	10.5	8.2–13.4	16.1	2.3	4.0	14.4	19.1	56.7	22.3
TEAC	13.0	10.5–15.6	12.1	2.4	2.5	12.0	12.2	97.2	24.4

Note: HT, days to heading from April 1; PH, plant height (cm); SY, seed yield (t ha^−1^); SW, specific weight (kg hL^−1^); TSW, 1000-seed weight (g); TPro, total proteins (%); TCarot, total carotenoids (μg β carotene g^−1^ dry matter); β-glu, β-glucans (%); TPh, total polyphenols (μg gallic acid equivalents (GAE) g^−1^ dry matter); TPant, total proanthocyanidins (μg catechin equivalents (CE) g^−1^ dry matter); TAnt, total anthocyanins (μg CE g^−1^ dry matter); TFlav, total flavonoids (μg gallic acid g^−1^ dry matter); DPPH (μmol of Trolox equivalent g^−1^ dry matter); TEAC (μmol of Trolox equivalent g^−1^ dry matter); CV, coefficient of variation; σ^2^_g_, genotypic variance; σ^2^_p_, phenotypic variance; GCV, genetic coefficient of variation; PCV, phenotypic coefficient of variation; h^2^_b_, broad-sense heritability; ΔG, expected response to selection at 5% of the selection intensity.

**Table 3 plants-10-01575-t003:** Phenotypic (above diagonal) and genotypic (below diagonal) coefficients of correlation (r) between the agronomic and qualitative traits for the 20 genotypes of barley.

Trait	HT	PH	SY	SW	TSW	TPro	TCarot	β-glu	TPh	TPAnt	TAnt	TFlav	DPPH	TEAC
HT		0.34	0.04	−0.44 *	0.13	−0.10	0.39	0.38	−0.21	0.63 **	−0.63 **	−0.27	−0.16	−0.10
PH	0.42		−0.59 **	−0.10	0.11	0.04	0.21	0.17	−0.31	0.15	−0.28	−0.30	−0.22	−0.25
SY	0.03	−0.77 ++		−0.09	0.10	−0.13	−0.06	0.34	0.47 *	0.22	0.12	0.25	0.52 *	0.46 *
SW	−0.48	−0.16	−0.12		0.37	0.23	−0.84 **	−0.13	−0.01	−0.31	0.43	0.08	0.15	0.12
TSW	0.18	0.10	0.10	0.37		−0.23	−0.25	0.15	−0.02	0.16	−0.17	0.09	0.25	0.01
TPro	−0.11	0.05	−0.13	0.23	−0.26		−0.12	0.39	0.22	−0.33	0.20	−0.20	0.19	0.40
TCarot	0.45	0.24	−0.08	−0.91 ++	−0.30	−0.12		0.25	−0.12	0.23	−0.54 *	0.01	−0.17	−0.19
β-glu	0.42	0.19	0.38	−0.14	0.18	0.39	0.26		0.29	0.27	−0.22	0.03	0.32	0.31
TPh	−0.23	−0.36	0.51	−0.01	−0.03	0.21	−0.10	0.28		0.35	0.18	0.17	0.81 **	0.91 **
TPAnt	0.71 +	0.19	0.24	−0.30	0.18	−0.32	0.21	0.26	0.36		−0.29	−0.02	0.28	0.32
TAnt	−0.68	−0.32	0.15	0.47	−0.17	0.21	−0.57	−0.23	0.18	−0.29		0.42	0.26	0.16
TFlav	−0.35	−0.35	0.30	0.09	0.16	−0.21	−0.01	0.06	0.19	−0.02	0.48		0.20	0.04
DPPH	−0.19	−0.31	0.65 ++	0.18	0.27	0.20	−0.20	0.32	0.92 ++	0.32	0.22	0.16		0.84 **
TEAC	−0.12	−0.29	0.52	0.12	0.02	0.41	−0.19	0.32	0.91 ++	0.33	0.15	0.03	0.95	

Note: HT, days to heading from April 1; PH, plant height (cm); SY, seed yield (t ha^−1^); SW, specific weight (kg hL^−1^); TSW, 1000-seed weight (g); TPro, total proteins (%); TCarot, total carotenoids (μg β carotene g^−1^ dry matter); β-glu, β-glucans (%); TPh, total polyphenols (μg gallic acid equivalents (GAE) g^−1^ dry matter); TPant, total proanthocyanidins (μg Catechin equivalents (CE) g^−1^ dry matter); TAnt, total anthocyanins (μg CE g^−1^ dry matter); TFlav, total flavonoids (μg gallic acid g^−1^ dry matter); DPPH (μmol of Trolox equivalent g^−1^ dry matter); TEAC (μmol of Trolox equivalent g^−1^ dry matter).* and ** Significance at *p* ˂ 0.05, *p* ˂ 0.01, respectively. Genotypic coefficients: significance of a genetic coefficient correlation is expressed as being greater than once (+) or twice (++) of its SE.

**Table 4 plants-10-01575-t004:** Eigenvalues, eigenvectors and percentage of variation explained by the first five principal components assessed for 14 traits in 20 barley genotypes evaluated at Foggia (Italy).

		Principal Component Axis			
	1	2	3	4	5
Eigenvalues	3.81	3.14	1.72	1.62	1.04
PC variation (%)	27.2	22.4	12.3	11.5	7.4
Cumulative variation (%)	27.2	49.6	61.9	73.5	80.9
**Trait**		**Eigenvectors**			
HT	−0.230	**0.402**	0.111	0.163	0.032
PH	−0.270	0.088	0.419	0.096	0.062
SY	0.307	0.208	−0.318	0.059	0.130
SW	0.208	−0.350	0.312	0.339	0.054
TSW	0.062	0.052	0.133	**0.672**	0.200
TPro	0.147	0.037	**0.520**	−0.427	0.236
TCarot	−0.252	0.336	−0.203	−0.298	0.196
β-glu	0.097	0.339	0.256	−0.045	**0.603**
TPh	**0.417**	0.225	0.012	−0.125	−0.194
TPAnt	0.026	**0.423**	−0.062	0.263	−0.261
TAnt	0.277	−0.331	−0.090	−0.097	0.127
TFlav	0.174	0.068	−0.415	0.093	**0.549**
DPPH	**0.425**	0.199	0.081	0.071	−0.080
TEAC	**0.420**	0.223	0.167	−0.118	−0.220

Note: HT, days to heading from April 1; PH, plant height (cm); SY, seed yield (t ha^−1^); SW, specific weight (kg hL^−1^); TSW, 1000-seed weight (g); TPro, total proteins (%); TCarot, total carotenoids (μg β carotene g^−1^ dry matter); β-glu, β-glucans (%); TPh, total polyphenols (μg gallic acid equivalents (GAE) g^−1^ dry matter); TPant, total proanthocyanidins (μg catechin equivalents (CE) g^−1^ dry matter); TAnt, total anthocyanins (μg CE g^−1^ dry matter); TFlav, total flavonoids (μg gallic acid g^−1^ dry matter); DPPH (μmol of Trolox equivalent g^−1^ dry matter); TEAC (μmol of Trolox equivalent g^−1^ dry matter). The bold values are the most important traits for each factor axes.

**Table 5 plants-10-01575-t005:** Number and means of genotypes with superior values for each of the 14 agronomic and biochemical traits, and superior lines for multiple traits, as identified by cluster analysis, with comparisons to parent values also given.

Trait	No. of Genotypes	Mean	Superior Line for Multiple Traits								Parent Comparison	
			L1997 (I)	L3004 (II)	L3005 (I)	L3006 (I)	L3007 (II)	L3008 (II)	L3009 (I)	Mean (M)	Parent Mean (PM)	Difference (%)
HT	11	11.1	-	11.6	10.4	11.8	11.6	-	11.0	11.3	13.6	−16.9
PH	5	89.9	-	-	88.9	89.4	91.1	-	90.0	89.9	102.5	−12.3
SY	4	5.2	-	-	5.4	5.3	-	-	-	5.4	4.5	+20.0
SW	1	82.1	-	-	-	-	-	-	-	82.1	77.5	+5.9
TSW	1	45.5	-	-	-	-	-	-	-	45.5	39.0	+16.7
TPro	3	14.3	-	-	-	-	14.0	14.3	-	14.2	12.7	+11.8
TCarot	2	3.9	-	-	-	-	-	-	-	3.9	3.1	+25.8
β-glu	2	4.6	-	-	-	-	-	-	-	4.6	4.5	+2.2
TPh	3	2780.7	2691.0	-	-	-	-	2734.0	2917.0	2780.7	2223.8	+25.0
TPAnt	3	1594.3	1630.0	-	-	-	-	-	-	1630.0	1407.3	+15.8
TAnt	2	128.2	124.0	-	-	-	-	-	132.3	128.1	31.8	+302.8
TFlav	3	1356.2	1421.7	-	-	-	-	-	1327.0	1374.4	979.7	+40.3
DPPH	6	12.7	12.4	12.8	12.6	-	12.4	-	13.4	12.7	9.7	+30.9
TEAC	4	15.2	-	14.8	14.7	-	-	15.6	15.6	15.2	12.2	+24.6

Note: HT, days to heading from April 1; PH, plant height (cm); SY, seed yield (t ha^−1^); SW, specific weight (kg hL^−1^); TSW, 1000-seed weight (g); TPro, total proteins (%); TCarot, total carotenoids (μg β carotene g^−1^ dry matter); β-glu, β-glucans (%); TPh, total polyphenols (μg gallic acid equivalents (GAE) g^−1^ dry matter); TPant, total proanthocyanidins (μg catechin equivalents (CE) g^−1^ dry matter); TAnt, total anthocyanins (μg CE g-1 dry matter); TFlav, total flavonoids (μg gallic acid g^−1^ dry matter); DPPH (μmol of Trolox equivalent g^−1^ dry matter); TEAC (μmol of Trolox equivalent g^−1^ dry matter). Roman numerals in parentheses refer to the corresponding cluster. M, mean of the lines with superior traits; PM, mean of 4 parental genotypes; Difference (%) = ((M − PM)/PM × 100). Superior trait for HT and PH: mean – σ; for all other traits: mean + σ.

## Data Availability

Not applicable.

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
