# Peer review of "Genetic Diversity for Agronomic Traits and Phytochemical Compounds in Coloured Naked Barley Lines"

_plants, 2021, doi:10.3390/plants10081575_

Round 1

Reviewer 1 Report

The manuscript is well written and the methods and results are nicely presented. I only have two suggestions/comments:

Lines 109-112: Why were the all of RILs considered as one group and all of the parents as another group? The RILs have different combinations of parents, and I assume that the parents are very distinct from each other (e.g. Ethiopian vs. Italian, hulled vs. hulless, black vs. white). This analysis is not statistically sound.

Lines 109-128: Trait acronyms need to be spelled out on first mention. For example, "...days to heading (HT), seed yield (SY)..."

Author Response

Responses to the comments of Reviewer 1:

Comments and Suggestions for Authors

The manuscript is well written and the methods and results are nicely presented. I only have two suggestions/comments:

Lines 109-112: Why were the all of RILs considered as one group and all of the parents as another group? The RILs have different combinations of parents, and I assume that the parents are very distinct from each other (e.g. Ethiopian vs. Italian, hulled vs. hulless, black vs. white). This analysis is not statistically sound.

Response: The sentence “Compared to the barley lines….. TEAC (+8%) (Table 2).” has been deleted. Consequently, the first two columns on the left of table 2 and the results in the abstract were also deleted (lines 7-8).

Lines 109-128: Trait acronyms need to be spelled out on first mention. For example, "...days to heading (HT), seed yield (SY)..."

Response: All acronyms have been spelled out on first mention.

The English language and style have been revised.

Reviewer 2 Report

The authors present evidence of useful genetic variation for potentially useful traits in nud barley breeding lines. The work is a significant contribution toward development of novel barley varieties and food products with high nutritional properties.

The manuscript is tidy but I identify some grammatical issues with special attention to the abstract:

line 9: "A barley breeding program was realized to obtain breeding material" is awkward in several ways. "was realized to obtain" does not have clear meaning. "breeding program...to obtain breeding material" is sort of redundant. I suggest something like "A barley breeding program was initiated to develop germplasm with naked and colored grains rich in bioactive compounds."

line 12: "experimental field" is like saying "experimental rocket". Is the field itself experimental. I suggest "experimental field trials" or "experimental trials"...the unless the field itself was experimental.

line 14: "traits and, to identify" I think the comma should be removed but I see how the authors are using the comma to avoid awkwardness using the workd "and" twice. Do you really need to say "agronomic performance" if you are assessing genetic variability for desired traits? maybe just say "aims to assess the genetic variability for desired traits and identify superior genotypes."  The next sentence provides more useful details.

line 18-19: "high genetic advance"...is not correct use of words. Are the authors referring to "selection response"? If so, I suggest "High heritability  and selection response was observed for most..." 

line 20: "(P  0.05)" missing symbol

line 482: "realized allowed to" does not make sense to me

There may be other issues in the manuscript, but I think the abstract must be improved.

Author Response

Responses to the comments of Reviewer 2:

Comments and Suggestions for Authors

The authors present evidence of useful genetic variation for potentially useful traits in nud barley breeding lines. The work is a significant contribution toward development of novel barley varieties and food products with high nutritional properties.

The manuscript is tidy but I identify some grammatical issues with special attention to the abstract:

line 9: "A barley breeding program was realized to obtain breeding material" is awkward in several ways. "was realized to obtain" does not have clear meaning. "breeding program...to obtain breeding material" is sort of redundant. I suggest something like "A barley breeding program was initiated to develop germplasm with naked and colored grains rich in bioactive compounds."

Response: The sentence has been replaced with the suggested one.

line 12: "experimental field" is like saying "experimental rocket". Is the field itself experimental. I suggest "experimental field trials" or "experimental trials"...the unless the field itself was experimental.

Response: It has been written "experimental trials".

line 14: "traits and, to identify" I think the comma should be removed but I see how the authors are using the comma to avoid awkwardness using the workd "and" twice. Do you really need to say "agronomic performance" if you are assessing genetic variability for desired traits? maybe just say "aims to assess the genetic variability for desired traits and identify superior genotypes."  The next sentence provides more useful details.

Response: The sentence has been replaced with the suggested one.

line 18-19: "high genetic advance"...is not correct use of words. Are the authors referring to "selection response"? If so, I suggest "High heritability  and selection response was observed for most..."

Response: The sentence has been replaced with the suggested one.

line 20: "(P  0.05)" missing symbol

Response: The symbol has been inserted.

line 482: "realized allowed to" does not make sense to me

Response: The sentence has been modified: “Our results indicate that it was possible to obtain, through a breeding program, barley lines….”

The English language and style have been revised.